# Determination of Risk Factors for Severe Life-Threatening Course of Multisystem Inflammatory Syndrome Associated with COVID-19 in Children

**DOI:** 10.3390/children10081366

**Published:** 2023-08-09

**Authors:** Ilia S. Avrusin, Natalia N. Abramova, Konstantin E. Belozerov, Gleb V. Kondratiev, Liudmila V. Bregel, Olesya S. Efremova, Alla A. Vilnits, Julia E. Konstantinova, Eugenia A. Isupova, Tatiana L. Kornishina, Vera V. Masalova, Eugeniy Yu. Felker, Olga V. Kalashnikova, Vyacheslav G. Chasnyk, Yuriy S. Aleksandrovich, Mikhail M. Kostik

**Affiliations:** 1Hospital Pediatry, Saint-Petersburg State Pediatric Medical University, Saint Petersburg 194100, Russia; 2Intensive Care Unite Department, Saint-Petersburg State Pediatric Medical University, Saint Petersburg 194100, Russia; 3Pediatric Oncology Department, Saint-Petersburg State Pediatric Medical University, Saint Petersburg 194100, Russia; 4Department of Pediatrics, Irkutsk State Medical Academy of Postgraduate Education, Branch of Russian Medical Academy of Continuous Professional Education, Irkutsk 664049, Russia; 5Department of Cardiology, Irkutsk Regional Children’s Hospital, Irkutsk 664022, Russia; 6Pediatric Infectious Department, Saint-Petersburg State Pediatric Medical University, Saint Petersburg 194100, Russia; 7The Research Department of Intensive Care of Emergency Conditions, Pediatric Research and Clinical Center for Infection Diseases, Saint Petersburg 197022, Russia; 8The Research Department of Vaccination and Adverse Event Follow Immunization, Pediatric Research and Clinical Center for Infection Diseases, Saint Petersburg 197022, Russia

**Keywords:** children, COVID-19, ICU, MIS-C

## Abstract

Multisystem inflammatory syndrome associated with COVID-19 in children (MIS-C) is a life-threatening condition that often requires intensive care unit (ICU) admission. The aim of this study was to determine risk factors for severe/life-threatening course of MIS-C. The study included 166 patients (99 boys, 67 girls) aged 4 months–17 years (median 8.2 years). The criterion of severity was the fact of ICU admission. To conduct a comparative analysis, MIS-C patients were divided into two groups: patients hospitalized in the ICU (*n* = 84, 50.6%) and those who did not need ICU admission (*n* = 82, 49.4%). Patients with a more severe course of MIS-C were significantly older. They had a higher frequency of signs such as rash, swelling, hepatomegaly, splenomegaly, and neurological and respiratory symptoms. Hypotension/shock and myocardial involvement were much more common in patients with severe MIS-C. These patients had a more significant increase in CRP, creatinine, troponin, and D-dimer levels. Additionally, the presence of macrophage activation syndrome was higher in patients admitted to the ICU. Conclusion: Nineteen predictors of severe course of MIS-C were found, out of which hepatomegaly, splenomegaly, D-dimer > 2568 ng/mL, troponin > 10 pg/mL were mainly associated with the probability of being classified as early predictors of severe MIS-C requiring ICU admission.

## 1. Introduction

In December 2019, the first case of a new coronavirus infection with severe acute respiratory syndrome (SARS-CoV-2) was registered in Wuhan, Hubei Province, China. Since then, the COVID-19 pandemic has rapidly escalated into a global health emergency worldwide. In children, the manifested form of this disease is less frequently observed than in adults and ranges from 1–5% to 18% of diagnosed diseases among the population, according to various researchers [1,2], and, as a rule, proceeds in an asymptomatic or mild form [3,4]. The multisystem inflammatory syndrome associated with COVID-19 in children (MIS-C) is a new challenging problem for pediatricians. MIS-C is a rare but serious disease associated with COVID-19 involving various systems and organs, including the heart, lungs, kidneys, brain, skin, eyes, and gastrointestinal tract. Initial descriptions of this condition began to appear in April 2020, a month after the start of the pandemic [5,6,7,8]. MIS-C is quite rare, accounting for less than 1% of children with confirmed SARS-CoV-2 infection. Its estimated incidence at the beginning of the pandemic in children was 2 cases per 100,000 people younger than 21 years of age in the USA [9]. According to more recent estimates, the incidence of MIS-C is 1:3000–4000 children who have had COVID-19 infection [10]. There are studies showing that the course of MIS-C seems to depend on race and ethnicity, with a disproportionately large number of cases among black and Latin American children, while among Asian children, this disease is less common [9,11,12,13]. Additionally, according to the CDC, the majority of MIS-C patients have been of Hispanic/Latino or non-Hispanic Black race/ethnicity. Hispanic/Latino and non-Hispanic Black populations are also disproportionately affected by COVID-19 overall [14]. It is noted that the onset of MIS-C occurs, as a rule, 2–6 weeks after COVID-19 [6,8,9]. Currently, the pathophysiology of MIS-C is insufficiently studied. It is assumed that the syndrome, as well as Kawasaki disease, is based on a violation of immune regulation with the development of an abnormal immune response to a viral pathogen with massive cytokine production [15,16]. It can also be associated with the prolonged persistence of SARS-CoV-2 in the gastrointestinal tract, leading to the release of zonulin, a biomarker of intestinal permeability, with subsequent trafficking of SARS-CoV-2 antigens into the bloodstream, leading to hyperinflammation [17]. There are also reports of the presence of autoantibodies in patients with MIS-C not only against endothelial but also against gastrointestinal and immunocompetent cells [18]. Some studies show that there are immune complexes consisting of antibodies to S-protein that cause macrophage activation [19,20].

MIS-C is rather difficult to diagnose since its symptoms do not differ significantly from the infectious process, and it also has a lot in common with Kawasaki disease (KD). So, in the first months after the pandemic started, such names as Kawa-COVID and Kawasaki-like syndrome were in use. In both diseases, fever, rash, conjunctivitis, erythema/edema of hands and feet, and cervical lymphadenopathy were observed [6,7,8,15], and some patients with MIS-C fulfilled the criteria of the American Heart Association (AHA) for KD [21]. Despite the presence of common clinical manifestations, MIS-C and KD are two different conditions [22,23]. MIS-C occurs in older children than in Kawasaki disease and is also often manifested by gastrointestinal symptoms (diarrhea, abdominal pain, vomiting) and heart damage (myocarditis, pericarditis), often leading to myocardial damage and shock, while these clinical manifestations occur in Kawasaki disease less often [24]. The course of MIS-C is significantly more severe, which is why patients need admission to the ICU in more than half of cases [22,23,24,25,26], and therapy is one of the most expensive among a variety of acute diseases.

The purpose of this study was to determine the risk factors for a severe life-threatening course of multisystem inflammatory syndrome associated with COVID-19 in children, which can be used as early predictors of ICU admission requirements.

## 2. Materials and Methods

### 2.1. Patients

The retrospective multi-central study included data from about 166 patients (99 boys, 67 girls), aged 4 months to 17 years (median 8.2 years) with a diagnosis of “multisystem inflammatory syndrome associated with COVID-19 in children” who were on inpatient treatment at the hospitals of St. Petersburg, Irkutsk, Yakutsk, Kaliningrad and other cities of Russia. All patients met the WHO criteria of MIS-C [27]. The study lasted from June 2020 to June 2022.

The history of previous COVID-19 infection in the studied patients was confirmed using at least one of the following methods: a positive result of PCR with reverse transcription (13%), the presence of antibodies of classes Ig M (40.3%) or Ig G (97.4%) to SARS-CoV-2, contact with a person with confirmed COVID-19 (65.6%).

The WHO criteria for MIS-C were used as the inclusion criteria for this study [27].


*Inclusion criteria (all items must be present):*
Age from 0 to 18 years;Fever for ≥3 days;Clinical signs of multisystem lesion (at least 2 of the following): Rash, bilateral non-purulent conjunctivitis, or signs of inflammation of the skin and mucous membranes (oral, hands, or feet);Hypotension or shock;Cardiac dysfunction, pericarditis, valvulitis, or coronary abnormalities (including echocardiographic data or elevated troponin/BNP);Signs of coagulopathy (prolonged PT or PTT; elevated D-dimer).Acute gastrointestinal symptoms (diarrhea, vomiting, or abdominal pain).Elevated inflammatory biomarkers (ESR, CRP, or procalcitonin)Signs of previous COVID-19: Any of the following: Positive PCR of SARS-CoV-2/Positive serological studies/Positive antigen test/Contact with a person with confirmed COVID-19.


### 2.2. Exclusion Criteria

Non-compliance with the inclusion criteria;The presence of an obvious microbial cause of multisystem inflammation, including bacterial sepsis and staphylococcal/streptococcal toxic shock syndrome.

### 2.3. Study Design

The frequency of clinical signs and the level of laboratory changes in patients with MIS-C were evaluated. We evaluated 80 parameters associated with MIS-C.

The parameters with the maximum deviation from the norm were taken into account.

A severe life-threatening course of MIS-C was determined as a requirement for ICU admission due to organ/system or multiorgan/multisystem failure, which required intensive care and monitoring. The main indications for ICU admission were shock/hypotension, severe-moderate respiratory disturbances, heart and CNS involvement, acute renal failure, and severe hematological abnormalities (predominantly thrombocytopenia with bleeding)

To conduct a comparative analysis, patients with MIS-C were divided into two groups: Group 1—patients admitted in the ICU (*n* = 84, 50.6%) and Group 2—patients who did not need ICU admission (*n* = 82, 49.4%).

Hemophagocytic syndrome was diagnosed based on HScore (Table 1), the presence of hemophagocytic lymphohistiocytosis (HLH) according to HLH-2004 criteria, or the presence of macrophage activation syndrome, according to 2005 and 2016 EULAR/ACR/PRINTO criteria for systemic juvenile idiopathic arthritis [28,29,30,31].

### 2.4. Statistics

The analysis of the obtained data was performed using the STATISTICA software package, version 10.0 (StatSoft Inc., St. Tulsa, OK, USA). The description of quantitative indicators is made with the indication of the median (25th; 75th percentiles). A comparison of qualitative indicators was carried out using the Pearson criterion χ2. A comparison of quantitative indicators was carried out using the Mann–Whitney criterion. The ability of each trait to differentiate patients with binary traits was evaluated using sensitivity and specificity analysis. For quantitative variables, cut-off values were calculated using AUC-ROC analysis (AUC—area under the curve—“area under the curve”) with the determination of 95% confidence interval (CI), calculation of the odds ratio (OR) without taking into account the time of development of events of interest using 2 × 2 tables. Sensitivity (Se) and specificity (Sp) were evaluated for each studied parameter. Independent predictors were established using binary logistic regression by including quantitative and qualitative indicators associated with the dependent variable in the analysis. Differences or connections were considered statistically significant at *p* < 0.05.

## 3. Results

### 3.1. General Characteristics of Patients with MIS-C

Male patients were affected more often (59.6%) than females; the median age of patients was 8 years, 2 months, minimum—4 months, maximum—17 years.

Most common clinical signs of MIS-C were as follows: fever (100%), conjunctivitis (84.8%), rash (78.9%), gastrointestinal symptoms (77.2%), cervical lymphadenopathy (66.9%), mucosal brightness (64%), hepatomegaly (64.4%), hands/feet erythema/edema (62.4%), sore throat (56.3%), face swelling (50.5%), respiratory symptoms (49.4%), red cracked lips (49.3%), neurological symptoms (47.8%), hypotension/shock (43.8%), splenomegaly (40.7%).

Among the laboratory parameters, most patients showed a high level of inflammatory markers, such as ESR (*n* = 150, 90.9%), CRP (*n* = 153/157, 97.5%), ferritin (*n* = 69/90, 76.7%), increased ALT (*n* = 85/164, 52.8%), AST (*n* = 101/148, 68.2%), LDH (*n* = 56/94, 59.6%), D-dimer (*n* = 120/125, 96.0%), and hypoalbuminemia (*n* = 115/182, 81.0%), hypoproteinemia (*n* = 102/126, 81.0%). Many patients had signs of hemophagocytosis, such as thrombocytopenia (*n* = 79, 47.6%), hyperferritinemia, increased liver enzymes, increased LDH and D-dimer, hypoalbuminemia, and hypoproteinemia.

Speaking of heart involvement, according to ECHO kg data, one-third of patients had changes characteristic of myocardial damage (30.6%) and pericardial effusion (28.8%), and 15.8% had CA dilation/aneurysms.

Patients with MIS-C received glucocorticosteroids (81.5%), acetylsalicylic acid (57.1%), intravenous immunoglobulin (44.7%, due to restricted access), and tocilizumab (4.9%). Replacement therapy with albumin, plasma preparations, erythrocyte mass, infusion therapy, and inotropic support were used as accompanying therapy.

The median length of stay in the hospital was 18 days.

There have been no lethal outcomes reported. All patients recovered.

### 3.2. Characteristics of Patients with Severe Life-Threatening Course of MIS-C

More than half of the patients (50.6%) were admitted to the intensive care unit. The main reasons for admission were systemic hemodynamic disorders (shock, arterial hypotension), severe respiratory disorders requiring oxygen support, and neurological disorders.

There were no differences in ways of COVID-19 identification between studied groups. Patients with a more severe course of MIS-C were relatively older (median—9.2 years) than patients who did not need the ICU admission (6.8 years, *p* = 0.006). Significant differences were noted in the frequency of signs such as rash (71.8% vs. 85.9%, *p* = 0.031), face swelling (62.5% vs. 41.3%, *p* = 0.027), neurological (57.5% vs. 38.3%, *p* = 0.015), and respiratory symptoms (58% vs. 40.7%, *p* = 0.028), hepatomegaly (78.1% vs. 49.3%, *p* = 0.0003), and splenomegaly (59.7% vs. 21.3%, *p* = 0.000002), compared with patients with a milder course of the disease. The most significant differences were found in the frequency of hypotension and shock, which were observed in severe MIS-C (75%) compared to milder—11% (*p* = 0.0000001). Signs of myocardial damage were more typical for severe MIS-C courses (43.9%) compared to milder patients—16.7% (*p* = 0.0002).

Patients with life-threatening course of MIS-C had anemia (hemoglobi *n* = 99 g/L vs. 110 g/L, *p* = 0.00001), leukocytosis (18.6 vs. 15.0 × 10^9^/L, *p* = 0.026), lower platelets (149 vs. 224 × 10^9^/L, *p* = 0.0006), high levels of CRP (178 mg/L vs. 101 mg/L, *p* = 0.0002), troponin (20.0 pg/mL vs. 2.7 pg/mL, *p* = 0.018), D-dimer (2224.5 ng/mL vs. 1064 ng/mL, *p* = 0.0002), and creatinine (589.5 mmol/L vs. 51.6 mmol/L, *p* = 0.027) and lower levels of protein (52.1 g/L vs. 58.6 g/L, *p* = 0.0001), albumin (28.0 g/L vs. 30.2 g/L (*p* = 0.009), fibrinogen (3.9 g/L vs. 5.2 g/L, *p* = 0.012) compared to patients with milder course. Patients admitted to the ICU frequently had signs of hemofagocytosis/macrophage activation syndrome (48.8% vs. 2.1%, *p* = 0.0003 according to the criteria for macrophage activation syndrome, 2005 [13]). Moreover, the Hscore was significantly higher among those hospitalized in the ICU compared with patients who did not need the ICU treatment (106 points vs. 75, *p* = 0.000007). Some differences were also found in treatment approaches. Thus, children hospitalized in the ICU were more likely to receive intravenous immunoglobulin therapy (53.7% vs. 35.1%, *p* = 0.018), and 7 out of 44 patients (15.9%) needed repeated IVIG courses (average dose 1 g/kg). Glucocorticosteroids were also used more often in ICU treatment (91.7% vs. 70.5%, *p* = 0.0005), mainly in the high-dose regimen (methylprednisolone 20–30 mg/kg 3–5 days in a single dose or divided into 3–4 equal doses). The median duration of hospitalization in patients treated in the ICU was 20 days compared to milder—16 days (*p* = 0.043). Patients admitted in the ICU met the criteria of Kawasaki disease, similar to patients with a milder course of MIS-C. The characteristics of patients are presented in Table 2.

The factors associated with ICU admission were identified based on indicators that had significant differences (Table 2), followed by an analysis of sensitivity and specificity and calculation of the odds ratio. The results of the univariate analysis are presented in Table 3.

Parameters with the highest sensitivity, specificity, odds ratio, and clinical significance were included in the multivariate regression analysis, excluding duplicate factors. Out of the initial 19 predictors included in the model, only three variables (splenomegaly, D-dimer > 2568 ng/mL, troponin > 10 pg/mL) were significantly associated with the probability of being classified as early predictors of severe MIS-C requiring hospitalization in the intensive care unit (Table 4).

Thus, patients having these signs should be classified as a high-risk group for a severe life-threatening course of MIS-C. Careful monitoring of vital functions should be carried out, and therapy should be prescribed to prevent the disease progression.

## 4. Discussion

In our study, we identified the most significant clinical and laboratory parameters associated with a life-threatening course of MIS-C. Possibly, its close monitoring might prevent or decrease the severity of the disease and improve the outcomes or modify treatment options.

Intensive and pronounced inflammation, as well as signs of hemophagocytosis, were associated with life-threatening disease course [6,7,8,11,32,33,34]. Special attention is focused on this pathology due to the rapid development and high frequency of life-threatening complications—shock and multiorgan failure, including heart damage and coagulopathy [7,8,9].

Involvement of the cardiovascular system took place in 50% of patients in our study. Similar figures appear in the articles of other researchers, where it is described with a frequency from 33 to 67% [35,36,37]. The main cardiovascular manifestations in MIS-C were shock, cardiac arrhythmias, myocardial and pericardial damage, and dilatation of the coronary arteries [38]. Thus, hypotension and shock are also commonly observed (32–76%) [11,26,32,33,34]. In our study, hypotension and shock occurred in 43.8% of patients and was much more common for patients with a severe course of MIS-C (75% vs. 11% in patients with a milder form, respectively). According to echocardiography, the signs of myocardial damage are observed in 30–40% of cases, and dilation/aneurysms of the coronary arteries in 8–24% [8,11,32,33,34,37,38]. Signs of heart damage are significantly more common in severe MIS-C—myocardial damage—up to 50–60%, and coronary artery lesions up to 50% in some studies [12,34]. Quite the same is shown in our study. Among all patients, myocardial involvement was found in 30.6% of cases, but in those who were admitted to the ICU, 43.9% had signs of myocardial involvement.

Abrams J.Y. and co-authors noted that the age over 5 years and certain laboratory markers, such as elevated levels of troponin, BNP, pro-BNP, ferritin, C-reactive protein, and D-dimer, may also be associated with life-threatening manifestations such as shock and depression of heart function [25]. There are certain similarities proposed in our study, such as D-dimer > 2568 ng/mL and troponin > 10 pg/mL. Additionally, we propose splenomegaly as an early predictor of severe MIS-C.

Hemophagocytic syndrome is one of the most severe conditions in the MIS-C course, with an incidence of 18–76%, according to various studies [25,39,40,41,42]. In our study, there was also a significantly increased frequency of secondary hemophagocytic syndrome (according to the H-score index) in ICU patients since this syndrome is accompanied primarily by manifestations of serious hematological, hepatic, and neurological dysfunction with the need for intensive therapeutic measures.

M. Cattalini and co-authors compared MIS-C and Kawasaki disease, also assessing the presence of hemophagocytic syndrome. They noted that patients with MIS-C had signs of secondary hemophagocytic lymphohistiocytosis (HLH) in 18.4%, while patients with KD—only in 1.2% [40].

All patients included in the study were treated following current recommendations [43]. According to ACR guidelines [44] and British national consensus [45], the first-line therapy is IVIG, as in Kawasaki disease.

In our study, children admitted to the ICU were treated with IVIG more often (53.7% vs. 35.1%, *p* = 0.018). Glucocorticosteroids were also used more often in the ICU department (91.7% vs. 70.5%, *p* = 0.0005). Additionally, some of the patients admitted to the ICU were effectively treated with tocilizumab. It is worth mentioning that there is no unified opinion about therapeutic tactics.

The question about the necessity of IVIG is still open [46]. Licciardi F. et al. prove that methylprednisolone mono-therapy as a first-line treatment is an effective option, especially in countries where IVIG is not available [46]. There are other studies comparing MIS-C therapy options. Thus, Ouldali N. and co-authors in their study showed that the combination of GCS and IVIG led to a significant reduction in the necessity of hemodynamic support and in the duration of stay in the ICU [47]. In another study, it was shown that combined therapy with IVIG and steroids leads to a decrease in the risk of cardiovascular dysfunction in patients with MIS-C, compared to patients who received only IVIG [48].

The initial treatment with glucocorticosteroids and supportive pathogenetic therapy (for shock and heart failure) is recommended for severe MIS-C, proceeding without signs of Kawasaki syndrome, and a combination of intravenous immunoglobulin and glucocorticosteroids with supportive therapy for MIS-C with signs of Kawasaki syndrome according to the current WHO recommendations [49]. However, these recommendations still have a low level of evidence base, and the problem of the optimal treatment protocol requires continued research [49]. We suggest that patients with predictors of severe course (including our predictors) might be candidates for early aggressive step-up treatment, which could modify the outcomes, with high-dose glucocorticosteroids and with intravenous anakinra in patients steroid-not responder [50].

### Study Limitations

The main study limitations are the possibility of bias in the selection of this population, partially missing data, relatively small sample size, different study time intervals, and different SARS-CoV-2 variants affected patients during the pandemic and the impossibility of having the laboratory data from the same time point. The decision to be admitted to the ICU is based on both one factor and a combination of factors, taking into account the patient’s age. The severity of each factor also influenced the decision to hospitalize the patient in the intensive care unit. Additionally, this decision could depend on the qualifications of the doctor and the capabilities of the hospital.

## 5. Conclusions

It is shown that more than half (50.6%) of patients need hospitalization in the ICU. The main factors determining the severity of MIS-C are as follows: damage to cardiovascular and central nervous systems and the presence of respiratory and hemodynamic disorders. It has been established that hepatomegaly, splenomegaly, D-dimer > 2568 ng/mL, and troponin > 10 pg/mL allow for the identification of a group of patients at risk of severe MIS-C who need hospitalization in the ICU. Patients with risk factors should be carefully monitored for vital functions and prescribed therapy to prevent the progression of the disease. We recommend paying special attention to the dynamics of D-dimer, troponin, platelets, and routine repeated evaluation of Hscore in every patient with suspicion of MIS-C.

## Figures and Tables

**Table 1 children-10-01366-t001:** The HScore [28].

Parameter	No. of Points (Criteria for Scoring)
Known underlying immunosuppression *	0 (no) or 18 (yes)
Temperature (°C)	0 (<38.4), 33 (38.4–39.4), or 49 (>39.4)
Organomegaly	0 (no), 23 (hepatomegaly or splenomegaly), or 38 (hepatomegaly and splenomegaly)
No. of cytopenias **	0 (1 lineage), 24 (2 lineages), or 34 (3 lineages)
Ferritin (ng/mL)	0 (<2000), 35 (2000–6000), or 50 (>6000)
Triglyceride (mmol/L)	0 (<1.5), 44 (1.5–4), or 64 (>4)
Fibrinogen (g/L)	0 (>2.5) or 30 (≤2.5)
Aspartate aminotransferase (IU/L)	0 (<30) or 19 (>30)
Hemophagocytosis features on bone marrow aspirate	0 (no) or 35 (yes)

* Human immunodeficiency virus-positive or receiving long-term immunosuppressive therapy (i.e., glucocorticoids, cyclosporine, azathioprine). ** Defined as a hemoglobin level of ≤9.2 g/dL and/or a leukocyte count of ≤5000/mm^3^ and/or a platelet count of ≤110,000/mm^3^.

**Table 2 children-10-01366-t002:** Comparison of characteristics of patients hospitalized in the ICU and patients who did not need hospitalization in the ICU.

Parameter	Total (*n* = 166)	ICU (*n* = 84)	No ICU (*n* = 82)	*p*-Value *
**Demographics**				
Age, months	98 (59; 134)	110 (66; 153)	84 (48; 129)	**0.006**
Gender, male, *n* (%)	99 (59.6)	51 (60.7)	48 (58.5)	0.775
**COVID-19 identification**				
PCR, *n* (%)	21/162 (13)	12/82 (14.6)	9/80 (11.3)	0.521
IgM, *n* (%)	52/129 (40.3)	28/57 (49.1)	24/72 (33.3)	0.069
IgG, *n* (%)	148/152 (97.4)	73/75 (97.3)	75/77 (97.4)	0.979
Family contact, *n* (%)	55/84 (65.5)	23/31 (74.2)	32/53 (60.4)	0.199
**Clinical signs**				
GI symptoms, *n* (%)	125/162 (77.2)	68/83 (81.9)	57/79 (72.2)	0.138
Neurological symptoms, *n* (%)	77/161 (47.8)	46/80 (57.5)	31/81 (38.3)	**0.015**
Sore throat, *n* (%)	89/158 (56.3)	41/78 (52.6)	48/80 (60)	0.346
Rash, *n* (%)	123/156 (78.9)	56/78 (71.8)	67/78 (85.9)	**0.031**
Conjunctivitis, *n* (%)	129/152 (84.9)	62/75 (82.7)	67/77 (87)	0.455
Dry cracked lips, *n* (%)	73/148 (49.3)	34/74 (46)	39/74 (52.7)	0.411
Bright mucous, *n* (%)	73/114 (64)	28/49 (57.1)	45/65 (69.2)	0.183
Respiratory signs, *n* (%)	80/162 (49.4)	47/81 (58)	33/81 (40.7)	**0.028**
Cervical lymphadenopathy, *n* (%)	10/154 (66.9)	51/76 (67.1)	52/78 (66.7)	0.954
Hands/feet erythema/edema, *n* (%)	93/149 (62.4)	43/73 (58.9)	50/76 (65.8)	0.386
Peeling of fingers, *n* (%)	51/143 (35.7)	26/70 (37.1	25/73 (34.3)	0.718
Face swelling, %	56/111 (50.5)	30/48 (62.5)	26/63 (41.3)	**0.027**
Hepatomegaly, *n* (%)	94/148 (63.5)	57/73 (78.1)	37/75 (49.3)	**0.0003**
Splenomegaly, *n* (%)	59/147 (40.1)	43/72 (59.7)	16/75 (21.3)	**0.000002**
Arthritis/Arthralgia, *n* (%)	22/150 (14.7)	10/73 (13.7)	12/77 (15.6)	0.744
Shock/hypotension, *n* (%)	72 (43.4)	63 (75)	9 (11)	**0.0000001**
Duration of fever, days	9 (7; 12)	8 (6; 11)	10 (7; 13)	0.064
KD criteria fulfillment,				0.089
complete, *n* (%)	72 (43.4)	31 (36.9)	41 (50)	
incomplete, *n* (%)	38 (22,9)	6 (7.1)	2 (2.4)	
not meet, *n* (%)	56 (33.7)	47 (56)	39 (47.6)	
**Laboratory**				
Red blood cells, 10^12^/L	3.8 (3.4; 4.2)	3.6 (3.3; 3.9)	4.0 (3.7; 4.5)	**0.0000001**
Hemoglobin, g/dL	10.4 (9.4; 11.4)	9.9 (9.1; 11.2)	11.0 (10.0; 11.9)	**0.00001**
White blood cells, 10^9^/L	16.4 (12.2; 21.8)	18.6 (13.5; 23.1)	15.0 (11.4; 20.2)	**0.026**
Platelets, 10^9^/L	178 (100; 451)	149 (82.5; 436.5)	224 (139; 537)	**0.0006**
ESR, mm/h	41 (28; 52)	41 (32; 50)	40 (25; 53)	0.541
C-reactive protein, mg/dL	13.8 (4.4; 23.6)	17.8 (11.1; 26.5)	10.1 (2.7; 18.9)	**0.0002**
Ferritin, μg/L	265 (102; 476)	330 (92; 680)	226 (126; 385)	0.303
ALT, IU/L	44 (25; 74)	47 (29; 82)	37 (21; 69)	0.057
AST, IU/L	51 (34; 0.0)	51 (34; 77	51 (34; 79)	0.632
Serum protein, g/L	56.5 (49.0; 63.0)	52.1 (45.0; 59.5)	58.6 (53.4; 67.0)	**0.0001**
Albumin, g/L	29.3 (25.8; 34.0)	28.0 (25.0; 32.0)	30.2 (27.4; 35.0)	**0.009**
Triglycerides, mmol/L	2.3 (1.7; 3.2)	2.3 (1.7; 3.4)	2.3 (1.7; 2.7)	0.755
Creatinin, mmol/L	55.2 (42.6; 70.0)	59.5 (47.0; 76.9)	51.6 (40.4; 63.6)	**0.027**
LDH, IU/L	470 (273; 685)	498 (366; 687)	387 (261; 680	0.201
Fibrinogen, g/L	4.5 (2.6; 6.5)	3.9 (1.9; 5.9)	5.2 (3.1; 7.4)	**0.012**
D-dimer, ng/mL	1800 (751; 3275)	2225 (1030; 3965)	1064 (586; 2200)	**0.0002**
Troponin, pg/mL	5.1 (1.0; 56.0)	20.0 (5.0; 122.0)	2.7 (0.3; 9.8)	**0.018**
Hscore	91 (68; 132)	106 (90; 145)	75 (60; 112)	**0.000007**
HLH-2004, *n* (%)	9/157 (5.7)	7/81 (8.6)	2/76 (2.6)	0.105
MAS 2005, *n* (%)	56/158 (35.4)	40/82 (48.8)	16/76 (21.1)	**0.0003**
MAS 2016, *n* (%)	18/157 (11.5)	12/81 (14.8)	6/76 (7.9)	0.174
**Echo findings**				
CA dilatation/aneurism, *n* (%)	25/158 (15.8)	9/79 (11.4)	16/79 (20.3)	0.127
Myocardial involvement, *n* (%)	49/160 (30.6)	36/82 (43.9)	13/78 (16.7)	**0.0002**
Pericarditis, *n* (%)	46/160 (28.8)	27/81 (33.3)	19/79 (24.1)	0.194
**Treatment and outcomes**				
IVIG treatment, *n* (%)	71/159 (44.7)	44/82 (53.7)	27/77 (35.1)	**0.018**
IVIG 2nd dose, *n* (%)	7/71 (9.9)	7/44 (15.9)	0 (0)	**0.028**
Acetylsalicylic acid, *n* (%)	84 (50.6)	34 (40.5)	50/74 (61.0)	**0.009**
Glucocorticosteroids, *n* (%)	132/162 (81.5)	77 (91.7)	55/78 (70.5)	**0.0005**
Biologics, *n* (%)	6 (3.6)	4 (4.8)	2 (2.4)	0.423
Stay in hospital, days	18 (13; 25)	20 (14; 27)	16 (12; 21)	**0.043**

**Abbreviations:** GI—gastrointestinal; ICU—intensive care unit; ESR—erythrocyte sedimentation rate; ALT—alanine aminotransferase; AST—aspartate aminotransferase; LDH—lactate dehydrogenase; IVIG—intravenous immunoglobulin; CA—coronary artery; MI—myocardial involvement; *—comparison between admitted and not admitted in the ICU.

**Table 3 children-10-01366-t003:** Parameters associated with a severe course of MIS-C requiring hospitalization in the ICU.

Parameter	Se	Sp	AUC	OR (95%CI)	*p*
Age > 97.0 months	59.5	59.3	0.623 (0.544; 0.697)	2.1 (1.2; 4.0)	0.016
Rash	71.8	14.1	-	0.41 (0.2; 0.94)	0.031
Respiratory signs	58.0	59.3	-	2.0 (1.1; 3.8)	0.028
Face swelling	62.5	58.7	-	2.4 (1.1; 5.1)	0.027
Hepatomegaly	78.1	50.7	-	3.7 (1.8; 7.5)	0.0003
Splenomegaly	59.7	78.7	-	5.5 (2.7; 11.3)	0.000002
Shock/hypotension	75.0	89.0	-	24.3 (10.4; 57.0)	0.000000
Hb ≤ 9.9, gd/L	58.0	78.2	0.701 (0.624; 0.771)	4.6 (2.3; 9.2)	0.000007
WBC > 16.4 × 10^9^/L	61.9	66.2	0.622 (0.543; 0.696)	3.9 (2.0; 7.5)	0.00004
PLT ≤ 114 × 10^9^/L	44.0	87.5	0.658 (0.580; 0.731)	5.5 (2.5; 12.2)	0.000008
CRP >12.0, mg/dL	73.2	54.2	0.647 (0.587; 0.741)	3.3 (1.7; 6.5)	0.0004
Ferritin > 314.0, μg/L	52.9	71.2	0.551 (0.458; 0.641)	2.7 (1.3; 5.8)	0.01
Albumin ≤ 27.2, g/L	48.1	75.0	0.615 (0.529; 0.695)	2.7 (1.3; 5.5)	0.006
Fibrinogen ≤ 2.4 g/L	34.6	81.2	0.620 (0.536; 0.699)	4.2 (1.8; 10.0)	0.0007
D-dimer > 2568, ng/mL	45.6	84.2	0.694 (0.606; 0.774)	4.5 (1.9; 10.5)	0.0004
Troponin > 10, pg/mL	52.2	84.6	0.699 (0.551; 0.822)	6.0 (1.6; 23.0)	0.006
Hscore > 91	68.3	69.7	0.707 (0.630; 0.777)	5.0 (2.5; 9.8)	0.000002
MAS 2005	48.8	78.9	-	3.6 (1.8; 7.2)	0.0003
Myocardial involvement	43.9	83.3	-	3.9 (1.9; 8.2)	0.0002

**Abbreviations:** Se—sensitivity; Sp—specificity; OR—odds ratio; CI—confidence interval; WBC—white blood cells; PLT—platelets; CRP—C-reactive protein; MAS—macrophage activation syndrome.

**Table 4 children-10-01366-t004:** Early predictors of a severe course of MIS-C requiring hospitalization in the ICU.

Parameter	Β	SE	*p*-Value
Splenomegaly	0.30	0.128	0.02
D-dimer > 2568 ng/mL	0.45	0.120	0.0006
Troponin > 10 pg/mL	0.33	0.111	0.005

## Data Availability

The datasets generated during and/or analyzed during the current study are available from the corresponding author upon reasonable request.

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
