# Peer review of "Determination of Risk Factors for Severe Life-Threatening Course of Multisystem Inflammatory Syndrome Associated with COVID-19 in Children"

_children, 2023, doi:10.3390/children10081366_

Round 1

Reviewer 1 Report

The MIS-C definition is already well described, because about this condition is reported in pediatric studies from different countries.

The diagnostic criteria include clinical, laboratory, and other data for patients. Children who develop MIS-C have inflammation in specific organs and tissues such as the lungs, heart, kidneys, blood vessels, digestive system, brain, eyes and, skin. In addition, many immunological alterations have also been described. 

The title is informative enough to draw its potential reader's attention.

In the introduction/background section of this article, is not so successfully explained why current research is important. But the authors could provide some small additional information related to the problem and include additional relevant references.

Here are some suggestions that would improve the introduction:

1.           According to the context in the Introduction, the authors can give a scheme/figure showing pathogenesis of MIS-C, CDC criteria for the diagnosis of MIS-C, involvement of organ systems, etc.

2.           You can give some information about if there is a relationship between demographic, social characteristics and ethnicity, and the MIS-C?

Line 48 - According to CDC, MIS-C occurred in 1 of approximately 3,000 to 4,000 (https://www.cdc.gov/mis/mis-c/hcp/provider-families.html)

ref.2 - N/A ()

In chapter “Materials and Methods” - give the period/time range of the studied patients.

line 137 - "......most patients showed a high level of inflammatory markers, such as ESR, CRP, ferritin, increased ALT, AST, LDH, D-dimer....." .How many of all 166 patients showed a change in these markers (in one, two, in at least three...) as it is some of them that you are investigating for possible predictors of MIS-C?

In chapter “Results”, according to the context the author could add how many parameters they have studied in total and make the statistically significant parameters in the tables with different color (red/bold..) to distinguish the data clearly.

Line 256 - remove the question and paraphrase

Line – 265 – ref.

Line 269 – ref.

In “Study limitation” - necessity of a larger cohort of MIS-C patients, study time interval, with which waves/SARS-CoV-2 variants the study matches

The “Conclusions” chapter can be supplemented by authors’ proposals and recommendations for routine laboratory testing of certain parameters in children with COVID-19.

The manuscript is presented in an appropriate way, but can be  supplemented with more information and references.

Minor editing of English language required

Author Response

Dear Reviewer! Thank you so much for your positive evaluation of our manuscript. Our answers (A) on your queries (Q) are below and highlighted by green color in the manuscript.

The MIS-C definition is already well described, because about this condition is reported in pediatric studies from different countries.

The diagnostic criteria include clinical, laboratory, and other data for patients. Children who develop MIS-C have inflammation in specific organs and tissues such as the lungs, heart, kidneys, blood vessels, digestive system, brain, eyes and, skin. In addition, many immunological alterations have also been described. 

The title is informative enough to draw its potential reader's attention.

In the introduction/background section of this article, is not so successfully explained why current research is important. But the authors could provide some small additional information related to the problem and include additional relevant references. Here are some suggestions that would improve the introduction:

Q1. According to the context in the Introduction, the authors can give a scheme/figure showing pathogenesis of MIS-C, CDC criteria for the diagnosis of MIS-C, involvement of organ systems, etc.

A1. Dear Reviewer! The figure showing the pathogenesis added.

Q2. You can give some information about if there is a relationship between demographic, social characteristics and ethnicity, and the MIS-C?

A2. Dear Reviewer! The information about demographic, social characteristics and ethnicity added.

Q3. Line 48 - According to CDC, MIS-C occurred in 1 of approximately 3,000 to 4,000 (https://www.cdc.gov/mis/mis-c/hcp/provider-families.html)

A3. The information updated with recommended reference.

Q4. ref.2 - N/A ()

A4. Dear Reviewer! The references changed to more relevant. The list of references has been updated now.

Q5. In chapter “Materials and Methods” - give the period/time range of the studied patients.

A5. Dear Reviewer! The studied time interval lasted since June 2020 to June 2022

Q6. line 137 - "......most patients showed a high level of inflammatory markers, such as ESR, CRP, ferritin, increased ALT, AST, LDH, D-dimer....." .How many of all 166 patients showed a change in these markers (in one, two, in at least three...) as it is some of them that you are investigating for possible predictors of MIS-C?

A6. Dear Reviewer! The information was added.

Q7. In chapter “Results”, according to the context the author could add how many parameters they have studied in total and make the statistically significant parameters in the tables with different color (red/bold..) to distinguish the data clearly.

A7. Dear Reviewer! We evaluated 80 parameters. The information about the number of studied parameters added in the Methods section, Study design subsection.The p-value of all statistically significant parameters have been marked with bold.

Q8. Line 256 - remove the question and paraphrase

A8. The question removed and the sentence re-write.

Q9. Line – 265 – ref.

A10. The reference was added

Q10. Line 269 – ref.

A10. The reference was added

Q11. In “Study limitation” - necessity of a larger cohort of MIS-C patients, study time interval, with which waves/SARS-CoV-2 variants the study matches

A11. Dear Reviewer! The required information was added to the Limitation subsection.

Q12. The “Conclusions” chapter can be supplemented by authors’ proposals and recommendations for routine laboratory testing of certain parameters in children with COVID-19.

A12. Dear Reviewer! We recommend to take special attention to dynamics of D-dimer, troponin, platelets and routine repeated evaluation of Hscore in every patient with suspicion of MIS-C. The information added to the conclusion.

Q13. The manuscript is presented in an appropriate way, but can be supplemented with more information and references.

A14. Dear Reviewer! The additional information and references were added according your suggestions.

Q14. Minor editing of English language required

A14. The English editing done.

Dear Reviewer! Thank you so much! I hope the manuscript has became better after your suggestions

On behalf of the Authors

Mikhail Kostik, MD, PhD, Professor

Reviewer 2 Report

General comments

This is an original article which aimed to find the early predictors of severe MIS-C requiring ICU admission. The authors investigated the clinical data with multivariate analysis and created the predictive model. There are some issues which should be addressed for the publication. First, the criteria for ICU admission is not clear. Second, the methodology of statistical analysis is not clear. Third, some of the interpretations of the results seem inappropriate.

Major comments

  1. Please describe the criteria for the ICU admission. The criteria should not include the variables in Table 1 when the authors compared the ICU and nonICU groups (i.e shock, respiratory symptoms, or neurological symptoms could be the criteria to admit into ICU. In that case, the comparison of 2 groups is not appropriate). This also affects the result in Table 2 and 3.

  2. The authors described in the method section (page 2, Line 69), the present study as “the retrospective-prospective”. This is confusing. Please explain.

  3. Please write the details of Hscore.

  4. Please divide the total numbers of patients (in Table 1) and the groups of ICU and non ICU as the authors only compared the 2 groups. In the present Table1, it seems like the authors compare the 3 groups.

  5. In page 4, Line 182, the authors described that “Patients admitted in the ICU rarely met the criteria of Kawasaki disease than patients with a milder course of MIS-C”. However, there is no significant difference in terms of Kawasaki criteria fulfillment in Table 1. Please change the sentences.

  6. Please describe the details of IVIG (dose) or steroid therapy (dose).

  7. The details of statistical analysis are not clear enough. It is needed to describe how to choose the variables for odds ratio or how to define the confounding factors  the adjust for multivariable analysis. Please explain.

  8. Corticosteroid is written in several terms. Please unify.

Quality of English is fine. Please correct some typos and unify some terms.

Author Response

Dear Reviewer! Thank you so much for your kind evaluation of our manuscript. Our answers (A) on your queries (Q) are below and highlighted by green color in the manuscript.

This is an original article which aimed to find the early predictors of severe MIS-C requiring ICU admission. The authors investigated the clinical data with multivariate analysis and created the predictive model. There are some issues which should be addressed for the publication. First, the criteria for ICU admission is not clear. Second, the methodology of statistical analysis is not clear. Third, some of the interpretations of the results seem inappropriate.

Major comments

Q1. Please describe the criteria for the ICU admission. The criteria should not include the variables in Table 1 when the authors compared the ICU and nonICU groups (i.e shock, respiratory symptoms, or neurological symptoms could be the criteria to admit into ICU. In that case, the comparison of 2 groups is not appropriate). This also affects the result in Table 2 and 3.

A1. The decision to be admitted to the ICU is based on both one factor and a combination of factors, taking into account the patient's age. The severity of each factor also influenced the decision to hospitalize the patient in the intensive care unit. Also, this decision could depend on the qualifications of the doctor and the capabilities of the hospital.

 This also added in the Limitation subsection. The main indications were: shock/hypotension, respiratory disturbances,, heart and CNS involvement, acute renal failure and severe hematological abnormalities (predominantly thrombocytopenia with bleeding)

Q2. The authors described in the method section (page 2, Line 69), the present study as “the retrospective-prospective”. This is confusing. Please explain. Please write the details of Hscore.

A2. We change to retrospective. It is correct. The details of Hscore added.

Q3. Please divide the total numbers of patients (in Table 1) and the groups of ICU and non ICU as the authors only compared the 2 groups. In the present Table1, it seems like the authors compare the 3 groups.

A3. Dear Reviewer! P-value related to last two columns (admitted and not admitted in the ICU). I add aterics near the p-value and add explanation in the footnotes.

Q4. In page 4, Line 182, the authors described that “Patients admitted in the ICU rarely met the criteria of Kawasaki disease than patients with a milder course of MIS-C”. However, there is no significant difference in terms of Kawasaki criteria fulfillment in Table 1. Please change the sentences.

A4. Dear Reviewer! The sentence fixed according to table 1.

Q5. Please describe the details of IVIG (dose) or steroid therapy (dose).

A5. Dear Reviewer! The information added. The IVIG was near 1 g/kg, due to restricted access in the pandemy)

Q6. The details of statistical analysis are not clear enough. It is needed to describe how to choose the variables for odds ratio or how to define the confounding factors  the adjust for multivariable analysis. Please explain.

A6. Dear Reviewer! Firstly, we did univariate analysis, and all categorical variables were selected if the p-value were less 0.05. For quantitative variables we calculated cut-off with AUC-ROC analysis and selected the best sensitivity and specificity. Than we transform quantitative data to categorical and selected if p-value was less 0.05.

Than we calculated OR, Se, Sp for selected categorical variables (p<0.05).

All significant variables with high OR (excluding duplicated variables) were included in the multiple regression analysis and significant variables were extracted. We calculated points by βx100 and check the model with AUC-ROC analysis, sensitivity and specificity

Q7. Corticosteroid is written in several terms. Please unify.

A7. Dear Reviewer! We unified to glucocorticosteroids.

Q8. Quality of English is fine. Please correct some typos and unify some terms.                       A8. Dear Reviewer! Thank you. The typos were fixed and terms were unified.

Dear Reviewer! Thank you so much! I hope the manuscript has became better after your suggestions

On behalf of the Authors

Mikhail Kostik, MD, PhD, Professor

Round 2

Reviewer 1 Report

Dear Authors,

I have been carefully reviewed your revised article with the id number"children-2485155 ". In my opinion, this revised article incorporates all of the points raised in the original draft to the best of my knowledge.
The title is informative and relevant. The aim is also stated clear.
The references are relevant and recent. The cited sources are referenced correctly. Appropriate studies are included.The results is presented in an appropriate way.
Tables and figures are made appropriately.
Best wishes to all of the authors who contributed to the production of this wonderful work and congratulations on their future endeavors.

Author Response

Dear Reviewer!

Thank you so much for your positive evaluation of our revised manuscript.

We are fully confident that the manuscript has become better thanks to your comments and help.

On behalf of the Authors

Mikhail Kostik, MD, PhD, Professor